# Exploring the Molecular Origin for the Long-Range Propagation of the Substrate Effect in Unentangled Poly(methyl methacrylate) Films

**DOI:** 10.3390/polym15244655

**Published:** 2023-12-09

**Authors:** Jianquan Xu, Xiaojin Guo, Hongkai Guo, Yizhi Zhang, Xinping Wang

**Affiliations:** Institute for School of Chemistry and Chemical Engineering, Key Laboratory of Surface & Interface Science of Polymer Materials of Zhejiang Province, Zhejiang Sci-Tech University, Hangzhou 310018, China; xiaojinguo166@163.com (X.G.); kai200012172022@163.com (H.G.); zyz4308@163.com (Y.Z.)

**Keywords:** polymer/substrate interface, long-range substrate effect, unentangled polymer system, poly(methyl methacrylate) film

## Abstract

The polymer/substrate interface plays a significant role in the dynamics of nanoconfined polymers because of its suppression on polymer mobility and its long-range propagation feature, while the molecular origin of the long-range substrate effect in unentangled polymer material is still ambiguous. Herein, we investigated the propagation distances of the substrate effect (*h**) by a fluorinated tracer-labeled method of two unentangled polymer films supported on silicon substrates: linear and ring poly(methyl methacrylate) films with relatively low molecular weights. The results indicate that the value of *h** has a molecular weight dependence of *h**∝*N* (*N* is the degree of polymerization) in the unentangled polymer films, while *h**∝*N*^1/2^ was presented as previously reported in the entangled films. A theoretical model, depending on the polymer/polymer intermolecular interaction, was proposed to describe the above long-range propagation behavior of the substrate effect and agrees with our experiment results very well. From the model, it revealed that the intermolecular friction determines the long-range propagation of the substrate effect in the unentangled system, but the intermolecular entanglement is the dominant role in entangled system. These results give us a deeper understanding of the long-range substrate effect.

## 1. Introduction

The polymer/substrate interface, where the mobility of the nearby polymer chains usually can be suppressed by chemical or physical interactions with the substrate [1,2,3,4], is considered as one of the primary causes of introducing dynamics perturbations in supported polymer nanofilms [5,6,7,8] and adjusting their physical properties, such as glass transition temperature [2,9], crystallization behavior [1,10], viscosity [11,12], and other pivotal properties. Significantly, a large number of studies have found that the suppressed dynamics effect of a substrate (i.e., the substrate effect) can propagate to a long-range distance of approximately tens of nanometers (about 10–25 *R*_g,0_; *R*_g,0_ is the unperturbed root-mean-square radius of gyration) [1,7,13,14,15,16,17] to form an interfacial region. In this region, the mobility of polymer chains was suppressed with different degrees until it recovered to bulk chain mobility. Nearer to the substrate, slower chain mobility was observed. The thickness of this interfacial region could be defined as the propagation distance of the substrate effect. Importantly, the proportion of the interfacial region increases rapidly with decreasing film thickness and then the local dynamics (or properties) of the interfacial region dominate the average dynamics (or properties) of the polymer film [18]. Compared to other confinement effects (e.g., the free surface effect and finite size effect), the substrate effect has more opportunities to permit the adjustment of the properties of polymers in a controllable manner [2,11,19]. Hence, understanding the origin of the substrate effect and the pathway by which the substrate effect is able to long-range propagate the interior of polymers are the key points to further modifying the properties of confined polymer materials.

Specifically, it is assumed that the long-range substrate effect was propagated by the chain entanglement among polymer chains [20,21,22,23], as shown in most of the previous experiments for entangled polymer systems. Experimental studies [24,25,26] and a theoretical simulation [27] have both revealed a mobility transition zone [9,23] formed by interpenetration and entanglement between the suppressed polymer chains near the substrate (or polymer brushes) and the upper free chains. Koga et al. [3,28] and Napolitano et al. [29] experimentally revealed that the neighboring substrate chains with loop conformations provide the possibility to hinder the motions of polymer chains nearby via entanglement. However, different research groups observed that the long-range substrate effect unexpectedly existed in the unentangled polymer systems as well by using experimental [11,30,31] and simulation [32] methods. For example, Tsui et al. [11] reported that the propagation distance of a silica substrate on unentangled random copolymer films of styrene (St) and 4-methoxystyrene (MeOS) is ~2.6 to 6 *R*_g,0_ with increasing MeOS concentrations. de Pablo et al. [32] demonstrated that the *T*_g_ of short-chain polymer films could be varied markedly by the substrate effect when the film thickness decreased below ~15 *R*_g,0_. Thus, the molecular origin of the long-range substrate effect in unentangled polymer systems is still ambiguous. Further exploration is required and a general theoretical framework to describe such long-range substrate effects for both unentangled and entangled systems is needed.

In our previous work [21], a novel method was developed to detect the propagation distance of the substrate effect by the migration of a fluorinated tracer-labeled polymer in the bottom layer of bilayer samples but not applied in the unentangled system and also not focused on the essence of why the substrate can propagate such a long distance. In this paper, we investigated the propagation distance (*h_l_**) of the substrate effect in linear poly(methyl methacrylate) (*l*-PMMA) films with different number-average molecular weights (*M*_n_) (including unentangled and entangled systems) on silicon substrates to explore the molecular origin of the long-range substrate effect. It was found that *h_l_** increased with the degree of polymerization (*N*), and *h_l_**∝*N* was unexpectedly observed in the unentangled polymer films, while *h_l_**∝*N*^1/2^ was presented in the entangled films, as previously reported [21]. Combined with a physical model based on the idea that there is a relationship between the dimensions of polymer chains and the degree of intermolecular interaction, we revealed that the intermolecular friction determines the long-range propagation of the substrate effect in unentangled polymer systems, while intermolecular entanglement is the dominant role in the entangled polymer system. The propagation distances (*h_r_**) of the substrate effect in another unentangled system of ring poly(methyl methacrylate) (*r*-PMMA) films with different *M*_n_ were further investigated. The results were in a good agreement with the predictions from the model, which checked the validity of this model and our findings.

## 2. Materials and Methods

### 2.1. Materials

Methyl methacrylate (MMA, 99%), toluene (anhydrous, 99.8%), 2-bromoisobutyryl bromide (BIBB, 98%), *N*,*N*,*N*′,*N*″,*N*″-pentamethyl diethylenetriamine (PMDETA, 99%), and triethylamine (TEA, 99%) were purchased from Sigma-Aldrich Co., Saint Louis, MI, USA. 2-perfluorooctylethyl methacrylate (FMA, 97%) and benzotrifluoride (BTF, 99%) were purchased from J&K Scientific Ltd., Shanghai, China. 3-(trimethylsilyl)-2-propyn-1-ol (TMS-C≡C-CH_2_OH, 98%), Copper(I) bromide (CuBr, 99%), Copper(II) bromide (CuBr_2_, 99%), and tetrabutylammonium fluoride (TBAF, 1.0 M in tetrahydrofuran) were acquired from Aladdin Chemical Co., Ltd., Shanghai, China. Prior to polymerization, MMA and FMA were purified by passing them through a basic alumina column (100–200 mesh) and dried over CaH_2_. BTF was also dried over CaH_2_ and then distilled under reduced pressure. CuBr was purified by an acetic acid aqueous solution. All the other reagents and solvents were analytical grade and used as received.

### 2.2. Synthesis of Fluorinated Group Labeled Linear and Ring PMMA

Fluorinated *l*- and *r*-PMMA samples tracer-labeled with the FMA groups (*l*- and *r*-PMMA*_N_*-*tr*-FMA*_m_*) and PMMA homopolymer (PMMA_3200_, *M*_n_ = 320.0 kg/mol, PDI = 1.18) were synthesized using atom transfer radical polymerization (ATRP) [21,33] and “click” chemistry [34]. Please see the detailed synthesis process and characterizations in the Appendix A (see the descriptions of synthesis process and Appendix A). The *l*-PMMA*_N_*-*tr*-FMA*_m_* samples with *M*_n_ = 5.6–92.0 kg/mol (PDI ≤ 1.19, 56 < the degree of polymerization (*N*) < 920) and the *r*-PMMA*_N_*-*tr*-FMA*_m_* samples with *M*_n_ = 5.6–22.5 kg/mol (PDI ≤ 1.21, 56 < *N* < 225) were prepared (see Appendix A). Furthermore, the number of FMA groups (*m*) introduced to each chain was 1~4, as detected by ^19^F NMR using an internal standard method with trifluorotoluene. It had been confirmed that such few fluorinated units will not perturb the mobility of the polymer chains [21]. The range of *M*_n_ for *l*-PMMA-*tr*-FMA samples was 5.6–92.0 kg/mol, including *M*_n_ values both higher and lower than the critical entanglement molecular weight (*M_c_*, is about 18.4 [35]–29.5 kg/mol [36], its degree of polymerization, *N_c_*, located in 184–295). For *r*-PMMA-*tr*-FMA, 5.6 ≤ *M*_n_ ≤ 22.5 kg/mol, which is considered to be an unentangled system in such a low *M*_n_ region [37].

### 2.3. Film Preparation

Silicon (100) wafers covered by a native oxide layer (SiO*_x_*, ~2 nm) were cut into 1.3 cm × 1.3 cm slides and employed as substrates. The slides were cleaned by a freshly prepared piranha solution (H_2_SO_4_:H_2_O_2_ (*v*:*v*) = 3:1) at 363 K for 30 min. The bottom layers of the *l*-PMMA*_N_*-*tr*-FMA*_m_* and *r*-PMMA*_N_*-*tr*-FMA*_m_* films with different thicknesses were obtained by spin-coating solutions of the polymers in toluene at various concentrations onto the silicon substrates. The films were subsequently dried with vacuum annealing at 413 K for 24 h as insurance for the complete enrichment of fluorinated groups at the film surface. The top PMMA_3200_ layer (thickness of about 50 ± 2 nm) was prepared by a water-casting method as reported [7,21]. The thickness of each layer was determined by an EP^4^SW ellipsometer (Accurion GmbH Co., Göttingen, Germany).

### 2.4. Water Contact Angle Measurement

Water contact angles (WCA) were measured using a Krüss DSA-10 CA goniometer (Hamburg, Germany). In each measurement, a ~2 μL drop of deionized water was dispensed onto the sample surface. After a waiting time of 5 s, the contact angle of the droplet was measured.

## 3. Results

### 3.1. The Measurement of the Propagation Distance of the Substrate Effect in Fluorinated PMMA Films

To evaluate the propagation distance of the substrate effect (*h**) in silicon-supported PMMA films, we used a bilayer film system to record the time taken for the migration of the fluorinated groups from the surface of the bottom layer to the surface of the top layer, as our previous work [21]. Figure 1 shows the schematic of the structure of a bilayer film. The bottom layer was the fluorinated PMMA layer. Since the surface energy of the fluorinated groups is much smaller than that of PMMA chains, the fluorinated groups would diffuse easily to the film surface when the polymer chains are allowed to migrate. The prepared films were annealed at 413 K (about 20 K higher than the bulk *T*_g_) for 24 h as insurance for the complete enrichment of fluorinated groups at the film surface [38]. After annealing, its water contact angle increased from ~98° to ~114° since the fluorinated groups are hydrophobic. In addition, we have tried to leach the annealed films by toluene several times according to the reported procedure [39] and found the residual thicknesses of adsorbed layer were all <1 nm. That is, the effect of the thickness of adsorbed layer on the value of *h** could be ignored. The top layer was a ~50 nm PMMA_3200_ layer, which provided a constant distance for the migration of the fluorinated groups and separated the bottom layer from the air for a long enough distance to eliminate the free surface effect. When the bilayer film was annealed at a given temperature (higher than the bulk *T*_g_), the fluorinated groups would diffuse from the surface of the bottom layer to the surface of the bilayer film and the time spent for this process is related to the chain mobility at the bottom layer surface. Since the enrichment of the fluorinated groups at the surface of the bilayer film makes it more hydrophobic, a simple water contact angle measurement can monitor such diffusion process.

Figure 2a shows the evolutions of the water contact angle with annealing time at 403 K on the surface of PMMA_3200_//*l*-PMMA_225_-*tr*-FMA_2_ bilayer films with various thicknesses of the bottom layer (*h*). As shown for *h* = 98.4 nm, it is clear that the value of the water contact angle remained constant at the beginning of the annealing, while gradually increasing due to the gathering of fluorinated groups at the bilayer surface (verified by X-ray photoelectron spectroscopy in our previous work [21]). That is, the time at which the contact angle begins to rise is the minimum time (t* = 660 s) taken for the fluorinated groups to migrate from the surface of bottom layer to the surface of the bilayer films. With the decrease in *h*, more time was needed for the fluorinated groups to diffuse to the surface of the bilayer films. When *h* decreased from 98.4 to 16.5 nm, t* increased distinctly from 660 to 2634 s. Considering the diffusion distance (~50 nm) for various *h* is the same, a larger t* leads to slower chain mobility. Then, t* could be considered as a scale to reflect the mobility of the polymer chains at the surface of the bottom layer. Figure 2b plots the value of log(t*) versus *h* for PMMA_3200_//*l*-PMMA_225_-*tr*-FMA_2_ bilayer films. It was found that log(t*) kept constant at *h* > 47.4 nm, which suggested that the chain mobility at the bottom layer surface was similar and not suppressed by the substrate. For *h* < 47.4 nm, log(t*) increased rapidly. That is, *h* = 47.4 nm is the critical thickness for the propagation distance of the long-range substrate effect, i.e., *h** for *l*-PMMA_225_-*tr*-FMA_2_ film, *h_l_** = 47.4 nm. By this method, the value of *h** for other linear polymer films with different *M*_n_ could be easily obtained, as for ring polymer films.

### 3.2. The Molecular Weight Dependence of the Propagation Distance (h_l_*) of Substrate Effect in Linear PMMA Films

Figure 3 plots a compilation of the variations of log(t*) as a function of *h* for several representative molecular weights for PMMA_3200_//*l*-PMMA*_N_*-*tr*-FMA*_m_* bilayer films. For clarity, the data for different molecular weights were vertically shifted one relative to the other. The plots were similar to that shown in Figure 2b. As we discussed above, in the large *h* region, the mobility of these polymer chains was not perturbed by the substrate so that log(t*) was relatively small and maintained a constant value, while log(t*) increased dramatically in the small *h* region since the substrate effect suppresses the mobility of the polymer chains. Hence, the propagation distance of the substrate effect in linear PMMA films (*h_l_**) could be obtained at the thickness where log(t*) begins to increase. In addition, it was obvious that *h_l_** increased with the increase in *M*_n_.

Figure 4 shows the evolution of *h_l_** as a function of the degree of polymerization (*N*) at 403 K. It is clearly shown that the evolutions of *h_l_** could be divided into two regions: for *N* ≤ 235, where *h_l_** grows linearly with *N* (i.e., *h_l_**∝*N*); while for *N* > 235, *h_l_** is in direct proportion to *N*^1/2^ (i.e., *h_l_**∝*N*^1/2^). Considering that the degree of polymerization of polymers with critical entanglement molecular weight, *N_c_*, is about 184 [35]–295 [36] for linear PMMA, we suppose that the physical criterion to separate these two kinds of molecular weight dependence is whether the polymer matrix is entangled or not. For the entangled systems, a series of reported results [1,7,13,14,15,16,17] have already shown that *h** is in direct proportion to *N*^1/2^ (or *R*_g,0_) (i.e., *h**∝*R*_g,0_∝*N*^1/2^). For example, Siretanu et al. [22] found that the substrate can control the polymer film surface mobility remotely by investigating the surface destabilization of polystyrene films and the threshold thickness for the substrate effect could be normalized by the mean-square end-to-end distance of the polymer, *R*_EE_, which is also in direct proportion to *N*^1/2^. However, for the unentangled systems, the molecular weight dependence of *h_l_**∝*N* had a lack of attention in the references, which provides a new opportunity to understand the long-range substrate effect more deeply. At least, we believed that the molecular origin of the propagation of the substrate effect in the unentangled system should be different from that of the entangled system and the polymer size is important in the propagation of the substrate effect. Based on all the results and considerations above, we suggest that a mechanism related to the polymer size and the connectivity of the polymer chains [22] is responsible for the long-range effect of the substrate, while the most important factor impacting the connectivity from one polymer chain to another is intermolecular interaction. That is, because of the increasing polymer/polymer interaction and polymer size, *h_l_** increases with increasing *N*.

### 3.3. Exploring the Molecular Origin for the Long-Range Effect of Substrate

We then considered a physical picture whereby the interactions and topological constraints imposed by the substrate suppress the mobility of adjacent polymer chains and then gradually propagate and decay far from the substrate by intermolecular interaction. That is, the long-range propagation of the polymer/substrate interactions generates the long-range substrate effect. The following model was predicated on the ideas that intermolecular interaction is the dominant role in determining the long-range propagation of the substrate effect and there is a correlation between the dimensions of polymer chains and their corresponding degree of intermolecular permeability to describe the intermolecular interaction. Basically, the larger the size of a polymer chain, the larger the pervaded volume it occupies and the greater number of other polymer chains it would encounter to generate stronger intermolecular interactions [40]. Here, the pervaded volume of a polymer chain (*V*_p_) can be approximatively described as
(1)$$V_{p}=AR_{g,0}^{3}$$

where *A* is a constant of order unity and *R*_g,0_ is the unperturbed root-mean-square radius of gyration. If one polymer chain contains *N* constitutional units, the number of polymer chains (*n*) that completely fulfill a volume *V*_p_ can be calculated as [41]
(2)$$n=\frac{V_{p}\rho N_{A}}{NM_{0}}$$

where *ρ* is the mass density, *N*_A_ is Avogadro’s number, and *M*_0_ is the molar mass of the constitutional unit.

For an unentangled polymer system, the molecular interaction force is generated by the molecular friction [42]. Let *f* be the segmental friction factor and consider the size factor of the polymer chains in the direction normal to the substrate; thus, based on Equations (1) and (2), the molecular interaction force (*F*_m_) could be described as follows:(3)$$F_{m}\propto \frac{nNf}{R_{g,0}}=\frac{AR_{g,0}^{2}N_{A}f}{M_{0}}$$

Also, it was reported that the scaling relation of the entropic force (*F*_en_) of cone-tethered polymers interacting with a planar surface with respect to the polymer chain to surface distance, *h*, is $F_{\mathrm{en}}\left(h\right)\propto k_{B}T/h$ [43,44] and this relationship is valid when *h* is smaller than ~150 σ [43] (σ is the monomer length; for PMMA σ = 0.69 nm [45]) in some cases. *k*_B_ is the Boltzmann constant and *T* is the temperature on the Kelvin temperature scale. In our experiments, the fluorinated group in the polymer chain could be regarded as the “cone” and the topological deformation of a polymer chain is also caused by the perturbation of the substrate and the “cone”. Hence, we borrowed this scaling relation in our analysis. The polymer/substrate interaction force at a given *h*, *F*_s_(*h*), is balanced with the entropic force, i.e., $F_{s}\left(h\right)=F_{\mathrm{en}}\left(h\right)\propto k_{B}T/h$. Since the polymer/polymer interaction force also plays an important role on *F*_s_(*h*), the *h*-dependent *F*_s_ could be described as
(4)$$F_{s}\left(h\right)\propto \frac{kF_{m}k_{B}T}{h}=\frac{kAR_{g,0}^{2}N_{A}fk_{B}T}{M_{0}h}$$

where *k* depends on the substrate/polymer interaction and is a constant at a given temperature. Assuming that when *F*_s_(*h*) is smaller than a constant of *a*, the substrate effect on polymer mobility can be ignored, and we then have
(5)$$\begin{array}{cccc} h^{*}\propto \frac{kAR_{g,0}^{2}N_{A}fk_{B}T}{aM_{0}}\propto R_{g,0}^{2}\propto N & & N<N_{c} & \end{array}$$

For an entangled system, we suggest that the threshold molecular interaction force to hinder the polymer chain motion is determined by entanglement. That is, *F*_m_ should be related to the average number of entanglements (*Z* = 2*N*/*N_c_*) per chain:(6)$$F_{m}\propto \frac{Z}{R_{g,0}}$$

From Equations (4) and (6), the formula of *h** for entangled systems was also deduced:(7)$$\begin{array}{cccc} h^{*}\propto \frac{kZk_{B}T}{aR_{g,0}}\propto R_{g,0}\propto N^{1/2} & & N>N_{c} & \end{array}$$

The theoretical predictions of Equations (5) and (7) are consistent with our experimental results (Figure 4): *h_l_** is in direct proportion to *N* in the unentangled region, while *h_l_** is proportional to *N*^1/2^ in the entangled region. Hence, we preliminarily suggested that the long-range substrate effect is propagated by intermolecular friction in the unentangled system, while the entanglements dominate this process in the entangled polymer system.

In order to further check the validity of this model, the propagation distances of the substrate effect in other unentangled polymer systems of *r*-PMMA films (*h_r_**) with various molecular weights were also detected. Figure 5a plots a compilation of the variations of log (t*) as a function of *h* for PMMA_3200_//*r*-PMMA*_N_*-*tr*-FMA*_m_* bilayer films. Also, the value of *h_r_** could be obtained at the thickness where log(t*) begins to increase. Figure 5b plots the evolution of *h_r_** as a function of *N* in the region of 56 ≤ *N* ≤ 225. It was found that *h_r_** also increased linearly with *N* and was the same as that for unentangled linear polymers. The above result was in a good agreement with the prediction of Equation (5).

We also compared the value of *h_l_**/*h_r_** predicted by the model and obtained by the experiments. Considering that the parameters of *k*, *A*, *f*, *M*_0_, and *a* are the same for *l-* and *r*-PMMA, the determined parameter of *h** is *R*_g,0_. It was reported that $R_{g,0}^{l}=\sqrt{N\sigma^{2}/6}$ for linear polymers, while $R_{g,0}^{r}=\sqrt{N\sigma^{2}/12}$ for ring polymers [46]. Based on Equation (5), $h_{l}^{*}/h_{r}^{*}={\left(R_{g,0}^{l}\right)}^{2}/{\left(R_{g,0}^{r}\right)}^{2}=2$ could be easily predicted. Correspondingly, we observed that the value of *h_l_**/*h_r_** is independent of *N* and has a value of approximately 2 in our experimental results, as shown in Figure 5b, which is in accordance with the predictions of the model. Moreover, we also plotted the evolution of $h_{l}^{*}/R_{g,0}^{l}$ (or $h_{r}^{*}/R_{g,0}^{r}$) as a function of $R_{g,0}^{l}$ (or $R_{g,0}^{r}$) at 403 K for *l-* and *r*-PMMA films supported on silicon substrates, as shown in Figure 6. It was clearly presented that there are two regions with increasing $R_{g,0}^{l}$: $h_{l}^{*}/R_{g,0}^{l}$ increases linearly when $R_{g,0}^{l}$ < 4.32 nm (i.e., *N* < 235), while $h_{l}^{*}/R_{g,0}^{l}$ keeps constant at $R_{g,0}^{l}$ > 4.32 nm (i.e., *N* > 235), which is consistent with the reported results [21,22]. It has been found that the constant of $h_{l}^{*}/R_{g,0}^{l}$ in the entangled region depends on polymer itself, polymer/substrate interaction, and temperature. With the increasing polymer/substrate interaction or decreasing temperature, the constant increases. Again, the boundary condition of these two regions is defined as whether or not the *l*-PMMA system is entangled. Correspondingly, for *r*-PMMA, the evolution of $h_{r}^{*}/R_{g,0}^{r}$ with $R_{g,0}^{r}$ (1.49 nm ≤ $R_{g,0}^{r}$ ≤ 2.99 nm when 56 ≤ *N* ≤ 225) is overlapped with that of *l*-PMMA in the unentangled region. We can anticipate that with increasing $R_{g,0}^{r}$, the evolution of $h_{r}^{*}/R_{g,0}^{r}$ would still coincide with that of *l*-PMMA. Hence, we can use Figure 6 to predict the value of *h** for polymer systems with different topological structures and forecast that entanglements among the ring polymer would occur when $R_{g,0}^{r}$ > 4.32 nm (i.e., *N* > 470). Matsushita et al. [37] have investigated the melt rheology of ring polystyrenes with ultrahigh purity and found that the zero-shear viscosity deviated from the Rouse chains at *N* > 2.5 *N_c_* (i.e., for PMMA, *N* needs to be at least higher than 460), where new intermolecular interactions, e.g., entanglements, have occurred. This result agrees well with our prediction and reflects again that our model is reasonable.

The results above presented an insight into the molecular origin of the propagation of the long-range substrate effect, especially for unentangled polymer systems. Due to the interaction between polymer segments and the substrate, the local segment dynamics were drastically suppressed. From the perspective of the force transfer mechanism, the polymer/substrate interaction would propagate by chain connectivity [47] in one polymer chain at first with little or no attenuation. However, such interactions would attenuate obviously from one chain to another though intermolecular friction in the region of *N* ≤ *N_c_*. Hence, after several chain sizes, such an attenuated substrate effect on chain dynamics could be ignored. With increasing molecular weight, the intermolecular friction among polymer chains enhances and the corresponding intermolecular attenuation decreases, leading to a larger *h**. That is, the propagation distance is determined by the strength of the polymer/substrate interaction and intermolecular friction. For the situation where *N* > *N_c_*, the polymer/substrate interaction is propagated by the intermolecular entanglements, whereby the entanglement number per chain and the size of the polymer coil determine the propagation distance.

## 4. Conclusions

In summary, the utmost propagation distances of the substrate effect of linear and ring PMMA films (*h_l_** and *h_r_**) with various degrees of polymerization (*N*) on silicon substrates were measured precisely by a fluorinated tracer-labeled method to explore the molecular origin of the long-range propagation of the substrate. It was found that the evolution of *h_l_** with *N* could be divided into two regions by the polymerization degree of the polymer with the critical entanglement molecular weight (*N_c_*) (i.e., whether the system is entangled or not): for *N* ≤ *N_c_*, *h_l_** increases linearly with *N*, while *h_l_** increases linearly with *N*^1/2^ for *N* > *N_c_*. For other unentangled systems of ring PMMA films, the propagation distances of the substrate effect (*h_r_**) also increase linearly with *N*, which is same as that for linear PMMA. Based on the idea that there is a correlation between the dimension of the polymer chains and the degree of intermolecular interaction, and the consideration that intermolecular friction and entanglement determine the intermolecular interaction in unentangled and entangled polymer systems, respectively, a theoretical model was proposed to describe the long-range propagation behavior of the substrate effect. From this model, it is predicted that $h^{*}\propto R_{g,0}^{2}\propto N$ for *N* ≤ *N_c_*, while $h^{*}\propto R_{g,0}^{}\propto N^{1/2}$ for *N* > *N_c_*, both of which agree well with our experimental results. For further examining the validity of this model, we compared the value of *h_l_**/*h_r_** predicted by the model and obtained by the experiments. It was predicted that $h_{l}^{*}/h_{r}^{*}={\left(R_{g,0}^{l}\right)}^{2}/{\left(R_{g,0}^{r}\right)}^{2}=2$ at the same *N* for the model, which was in a good agreement with the experimental results. Consequently, it revealed that the intermolecular friction determines the long-range propagation of the substrate effect in unentangled systems, but the intermolecular entanglement is the dominant role in entangled system. These results help us to better understand the long-range substrate effect and provide the possibility to control the properties of supported polymer films by adjusting the propagation behavior of the substrate effect.

## Figures and Tables

**Figure 1 polymers-15-04655-f001:**
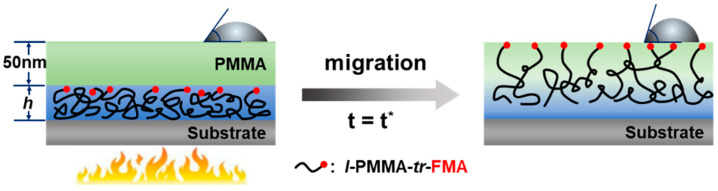
Schematic of the structure of a bilayer film and the migration of the fluorinated PMMA chains under annealing.

**Figure 2 polymers-15-04655-f002:**
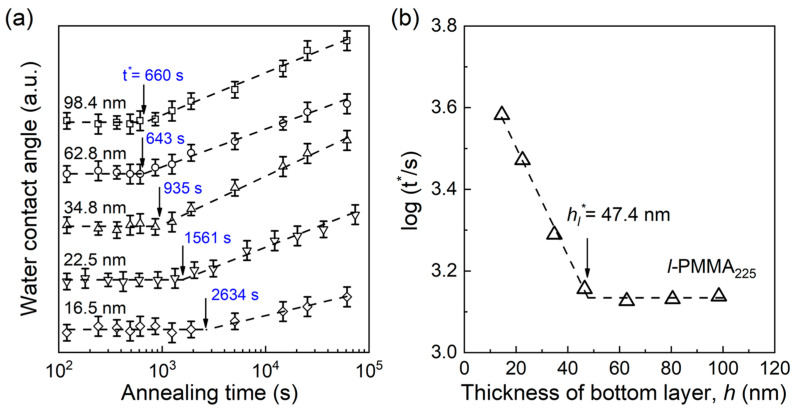
(**a**) The evolutions of water contact angles as a function of annealing time at 403 K on the surface of PMMA_3200_//*l*-PMMA_225_-*tr*-FMA_2_ bilayer films with typical thicknesses of the bottom layer (*h*). The results for different thickness films were vertically shifted one relative to the other for clarity. Usually, the water contact angle increased from ~74° to ~82° within our experimental time. (**b**) Plot of log (t*) (the unit of t* is second) versus the thickness of bottom layer (*h*) for PMMA_3200_//*l*-PMMA_225_-*tr*-FMA_2_ bilayer films. Dashed lines denote the best linear fits to the data to obtain the value of *h_l_**.

**Figure 3 polymers-15-04655-f003:**
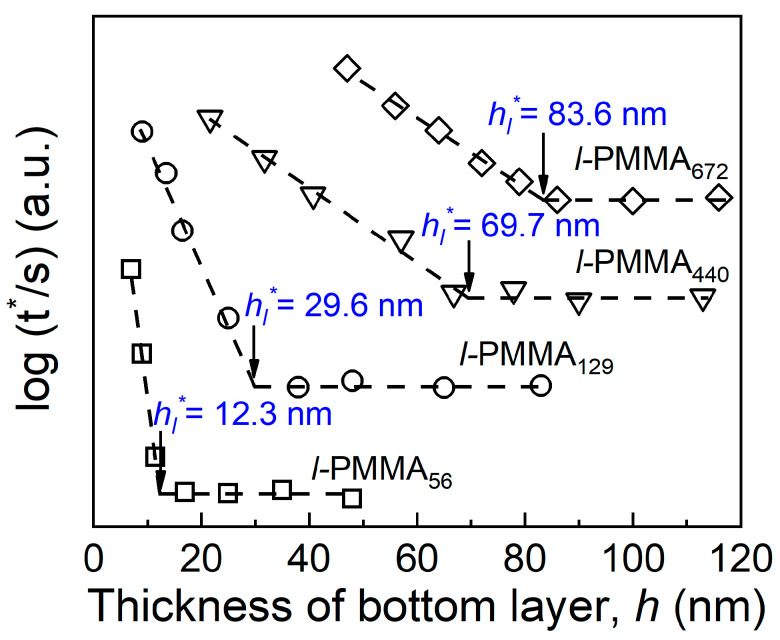
Plots of log(t*) (the unit of t* is second) versus the thickness of bottom layer (*h*) at several typical molecular weights for PMMA_3200_//*l*-PMMA*_N_*-*tr*-FMA*_m_* bilayer films. Dashed lines denote the best linear fits to the data to obtain the value of *h_l_**. The results for various molecular weights were vertically shifted one relative to the other for clarity. Usually, with the decreasing thickness, the value of log (t*) changes from 2.6–2.9 to 3.2–3.7 for different molecular weight linear polymer films.

**Figure 4 polymers-15-04655-f004:**
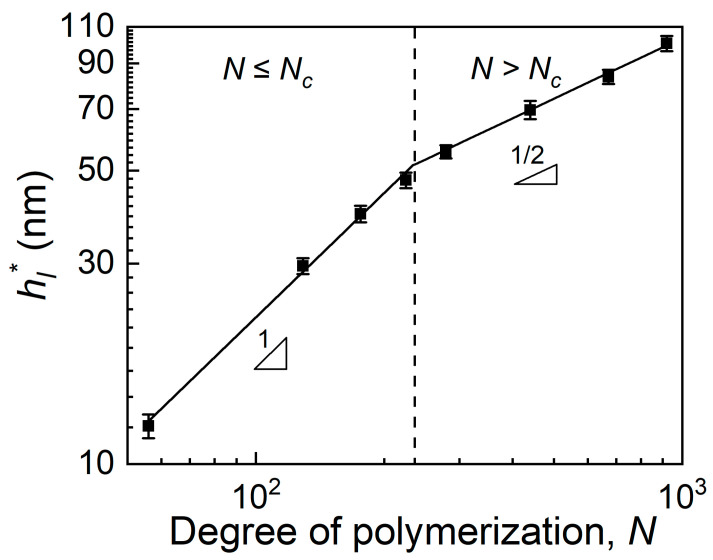
The evolution of *h_l_** as a function of the degree of polymerization (*N*) at 403 K. Solid lines denote the best linear fits to the data. *N_c_* is the polymerization degree of polymers with the critical entanglement molecular weight.

**Figure 5 polymers-15-04655-f005:**
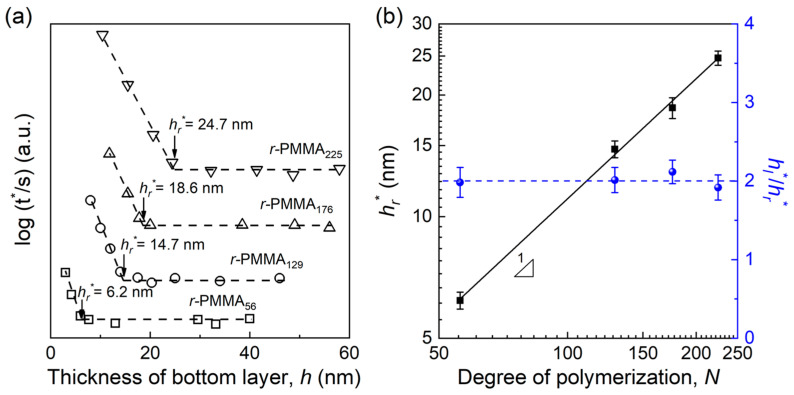
(**a**) Plots of log(t*) (the unit of t* is second) versus the thickness of bottom layer (*h*) at various molecular weights for PMMA_3200_//*r*-PMMA*_N_*-*tr*-FMA*_m_* bilayer films. Dashed lines denote the best linear fits to the data to obtain the value of *h_r_**. The results for various molecular weights were vertically shifted one relative to the other for clarity. Usually, with the decreasing thickness, the value of log(t*) changes from 3.2–3.5 to 3.8–4.5 for different molecular weight ring polymer films. (**b**) Evolutions of *h_r_** and *h_l_**/*h_r_**as a function of the degree of polymerization (*N*) at 403 K. Black solid line denotes the best linear fit to the data. Blue dashed line represents the line of *h_l_**/*h_r_** = 2.

**Figure 6 polymers-15-04655-f006:**
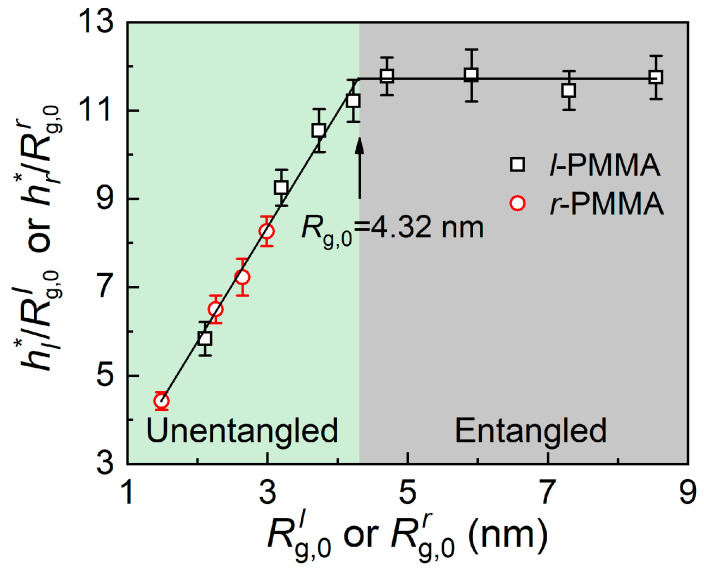
Plots of $h_{l}^{*}/R_{g,0}^{l}$ (or $h_{r}^{*}/R_{g,0}^{r}$) as a function of $R_{g,0}^{l}$ (or $R_{g,0}^{r}$) at 403 K for PMMA films supported on silicon substrate. Black lines are the best linear fitting to the data.

## Data Availability

The data presented in this study are available on request from the corresponding author.

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
