# Peer review of "Exploring the Molecular Origin for the Long-Range Propagation of the Substrate Effect in Unentangled Poly(methyl methacrylate) Films"

_polymers, 2023, doi:10.3390/polym15244655_

Round 1

Reviewer 1 Report

Comments and Suggestions for Authors

Publish as it is.

Comments on the Quality of English Language

Minor English language modification is necessary.

Author Response

Thank you very much for your suggestion of English language modification. We have examined the writing of the paper and tried our best to polish it. All the changes were colored in red in the revised manuscript.

Reviewer 2 Report

Comments and Suggestions for Authors

Comments to the Authors

I have gone through the manuscript entitles “Exploring the Molecular Origin for the Long-Range Propagation of the Substrate Effect in Unentangled Poly(methyl methacrylate) Films”. The authors have explained their work nicely. But, before acceptance, the authors need to address the following points and need minor revision of the manuscript.

1. In abstract section, within the sentence “linear and ring poly(methyl methacrylate) films with relative low molecular weights”, it is better to write “relatively” instead of “relative”.

2. In abstract section, the sentence “These results will help give us deeper insight to the long-range substrate effect in supported polymer films”, need to revise.

3. Similar type of language mistakes should be checked throughout the manuscript and need to correct accordingly.

4. Can entanglement be measured quantitatively?

5. Why hl* increases with the increasing of Mn? Need to explain.

6. Chain mobility is related to the glass transition temperature (Tg) of polymer. How can it be correlated with hl*?

7. The authors should mention how their work is different from the earlier publications.

8. In conclusion section, in the sentence “In summary, the umost propagation distances of the substrate effect of linear and ---------.” what is umost propagation? Or is this a spelling mistake?

Comments on the Quality of English Language

Quality of English is good. Minor revision required.

Author Response

Thank you very much for your helpful comments. We have examined them carefully and made a point-by-point response as follows. We hope that the revision of our manuscript meets the standard of Polymers and look forward to hearing good news from you.

Reviewer’s Comments & Author’s Responses

I have gone through the manuscript entitles “Exploring the Molecular Origin for the Long-Range Propagation of the Substrate Effect in Unentangled Poly(methyl methacrylate) Films”. The authors have explained their work nicely. But, before acceptance, the authors need to address the following points and need minor revision of the manuscript.

(1) In abstract section, within the sentence “linear and ring poly(methyl methacrylate) films with relative low molecular weights”, it is better to write “relatively” instead of “relative”.

Author Reply: Thanks for your kind reminder. The word of “relative” was revised to “relatively” and colored in red in the abstract section in the revised manuscript.

(2) In abstract section, the sentence “These results will help give us deeper insight to the long-range substrate effect in supported polymer films”, need to revise.

Author Reply: Thanks for your comments. The sentence “These results will help give us deeper insight to the long-range substrate effect in supported polymer films” was revised to “These results give us a deeper understanding to the long-range substrate effect.” in the last two lines of the abstract section in the revised manuscript.

(3) Similar type of language mistakes should be checked throughout the manuscript and need to correct accordingly.

Author Reply: Thanks for your kind comment. We have examined the writing of the paper carefully and tried our best to polish it. All the changes were colored in red in the revised manuscript.

(4) Can entanglement be measured quantitatively?

Author Reply: Thanks for your valuable comments. It is hard to measure the chain entanglement in the polymer thin film quantitatively, especially in substrate supported polymer film. To my best knowledge, there is only one method to detect the chain entanglement in the supported polymer thin film by following the time evolution of the wetting ridge at the surface induced by a droplet of ionic liquid as reported recently by our group (Macromolecules 2021, 54, 3735−3743). It was found that when the film thickness is larger than 3.3 Rg, the chain entanglement behavior in polymer film is similar with that of bulk material. In this paper, hl* is about 11.7 Rg  in the entangled systems (see Figure 6), which is much larger than 3.3 Rg. Hence, we think it is reasonable that we deal with the chain entanglement as same as the bulk in our theoretical model.

 (5) Why hl* increases with the increasing of Mn? Need to explain.

Author Reply: Thanks for your helpful comments. With the increasing of polymer molecular weight, the interaction of polymer/polymer and the size of polymer chain both increase, so that the substrate effect can propagate longer to show an increase in hl*. The sentence “That is, because of the increasing of polymer/polymer interaction and polymer size, hl* increases with the increasing of N.” was added in the lines 231-233 in the revised manuscript.

(6) Chain mobility is related to the glass transition temperature (Tg) of polymer. How can it be correlated with hl*?

Author Reply: Thanks for your helpful comments. hl* is the propagation distance of the substrate effect (i.e. the suppressed dynamics effect of a substrate). That is, the chain mobility in the region from substrate surface to a distance of hl* is suppressed by the substrate. If the value of hl* is large, the suppressed effect of the substrate on the chain mobility cannot be ignored. Hence, hl* is very important to the chain mobility for polymer thin film. In the previous work by our group (Phys. Rev. Lett. 2019, 122, 217801.), it was found that with the increasing of hl*, the thickness dependence of Tg changed a lot.

(7) The authors should mention how their work is different from the earlier publications.

Author Reply: Thanks for your kind reminder. The sentence “In our previous work [21], a novel method was developed to detect the propagation distance of the substrate effect by the migration of a fluorinated tracer-labeled polymer in the bottom layer of bilayer samples, but not applied in the unentangled system and also not focused on the essence of why the substrate can propagate so long distance.” was added in the lines 69-72 in the revised manuscript.

(8) In conclusion section, in the sentence “In summary, the umost propagation distances of the substrate effect of linear and ---------.” what is umost propagation? Or is this a spelling mistake?

Author Reply: Thanks for your valuable comments. It is a spelling mistake and the word should be “utmost”.

Reviewer 3 Report

Comments and Suggestions for Authors

In this manuscript, the authors report their investigation of long-range substrate effect in linear and ring PMMA film systems. This gives researchers a better understanding of polymer/substrate relationships. The manuscript is well-organized. It is recommended that this manuscript be published after minor revisions.

1.            The author should give a brief introduction of how the propagation is defined in this research and stress the importance of studying this parameter.

2.            Line 134, “Since the surface energy of fluorinated groups is much smaller than PMMA chains,”, please double check the grammar.

3.            In “Conclusion”, What does “umost” mean? Should it be “utmost”?

Author Response

Thank you very much for your helpful comments. We have examined them carefully and made a point-by-point response as follows. We hope that the revision of our manuscript meets the standard of Polymers and look forward to hearing good news from you.

Reviewer’s Comments & Author’s Responses

In this manuscript, the authors report their investigation of long-range substrate effect in linear and ring PMMA film systems. This gives researchers a better understanding of polymer/substrate relationships. The manuscript is well-organized. It is recommended that this manuscript be published after minor revisions.

(1) The author should give a brief introduction of how the propagation is defined in this research and stress the importance of studying this parameter.

Author Reply: Thanks for your kind reminder. The sentence “The thickness of this interfacial region could be defined as the propagation distance of the substrate effect.” and the words “until it recovers to bulk chain mobility” were added in the lines 39-41 in the revised manuscript. The importance of the propagation distance of the substrate effect was presented in the first paragraph of the Introduction section, please kindly find them.

(2) Line 134, “Since the surface energy of fluorinated groups is much smaller than PMMA chains,”, please double check the grammar.

Author Reply: Thanks for your kind reminder. The sentence was revised as “Since the surface energy of fluorinated groups is much smaller than that of PMMA chains” in the lines 142-143 in the revised manuscript.

(3) In “Conclusion”, What does “umost” mean? Should it be “utmost”?

Author Reply: Thanks for your valuable comments. It is a spelling mistake and the word should be “utmost”.